# Minimal Margin Surgery and Intraoperative Neuromonitoring in Benign Parotid Gland Tumors: Retrospective Clinical Study

**DOI:** 10.3390/jpm12101641

**Published:** 2022-10-03

**Authors:** Eva Aurora Massimilla, Giovanni Motta, Michelangelo Magaldi, Marco Montella, Gaetana Messina, Domenico Testa, Elena Cantone, Gaetano Motta

**Affiliations:** 1Department of Mental, Physical Health and Preventive Medicine-ENT Unit, University of Campania “L. Vanvitelli”, 80131 Naples, Italy; 2Department of Mental, Physical Health and Preventive Medicine-Pathology Unit, University of Campania “L. Vanvitelli”, 80138 Naples, Italy; 3Department of Cardio-Thoracic and Respiratory Sciences-Thoracic Surgery Unit, University of Campania “L. Vanvitelli”, 80131 Naples, Italy; 4Department of Neuroscience, Reproductive and Odontostomatological Sciences, Otorhinolaryngology-Head and Neck Surgery Unit, University of Naples Federico II, 80131 Naples, Italy

**Keywords:** extracapsular dissection, pleomorphic adenoma, Warthin tumor, intraoperative neuromonitoring, facial nerve

## Abstract

Extracapsular dissection (ECD) was introduced for the removal of superficial and small benign parotid tumors. According to a recent proposal, ECD is reserved for tumors that are 3 cm or less, mobile, and close to the parotid borders in cases of pleomorphic adenoma. The aim of the study is to evaluate the effectiveness of ECD for treatment of benign parotid tumors also in cases of tumors that were larger than 3 cm and deeper. All ECD for benign parotid neoplasms conducted between 2007 and 2017 were reviewed. The lesions included were limited to primary parotid tumors and categorized by Quer proposal. Facial nerve monitoring was used in all cases. Facial nerve palsy and local recurrences were assessed. The 88 ECD performed met inclusion criteria. The mean lesion size was 4.26 cm. Of the tumors, 68 were less than 3 cm in diameter and 20 were larger, 64 were superficial, and 24 were deep. The most common lesion types were pleomorphic adenoma (88.6%). There was no significant difference in complication rates between the size of tumor (*p* = 0.9) and location (*p* = 0.91). Our results suggest that extracapsular dissection could be considered an option for first-time diagnosed benign parotid tumors, even in cases of large dimensions and deep lobe involvement.

## 1. Introduction

Parotid gland tumors amount to just 1–3% of all primary head and neck neoplasms, of which 70% to 90% have benign histopathologic features [1,2,3,4,5,6].

Most of these benign lesions are pleomorphic adenomas (PA), followed by Warthin tumors (WT), and, less frequently, oncocytomas, basal cell adenomas, and others [4].

Although the treatment of these tumors is purely surgical, it has undergone several evolutionary steps over the past century. 

Before 1930, the primary goal of parotid surgery was to limit the risk of facial nerve palsy, so intracapsular enucleation (ICE) was the most common procedure performed [7].

However, the postoperative rate of recurrence with this technique was 20–40% [8,9].

The subsequent introduction of the superficial parotidectomy (SP) reduces the recurrence rate to about 2% [10], though it comes with a greater risk of complications such as facial paralysis, Frey syndrome (FS), salivary fistula, and aesthetic problems, due to major glandular resection [11,12].

Surgical techniques of “partial lobectomy” or “partial parotidectomy” [13,14,15,16] or “limited superficial parotidectomy’’ [17] emerged during the second half of the last century for treatment-limited tumors with the advantage of lower probability of facial paralysis and better aesthetic results [9]. 

For example, extracapsular dissection (ECD) with or without facial nerve dissection (ECD-FND) has emerged for removing superficial and small benign parotid tumors [7,8,9,11,12,18]. 

ECD can be differentiated from ICE, in that the latter consists of incising the tumor capsule and “shelling out” the neoplasm, with a high risk of relapse due to incomplete resection and the possible spread of the tumor in the surgical site [7]. 

ECD is performed by dissecting the tumor around the capsule with a 1–2 mm border of macroscopically healthy tissue but without pre-identification and dissection of the facial nerve trunk [10]. ECD with facial nerve dissection utilizes standard surgical landmarks to identify and dissect the facial nerve trunk early in the procedure [7,10,18].

In 2017, Quer et al. [9] considered several parameters, such as size, site, and histopathology, to decide the extent of surgery in the treatment of benign parotid tumors and proposed establishing a four-category classification by choosing a cut-off of 3 cm in diameter. Thus, their classification is based on size and site (levels according to the ESGS proposal [8]). They propose: category I for tumor ≤ 3 cm, superficial (outer surface), mobile and close to the parotid borders; category II for tumor ≤ 3 cm, deep, or far from the parotid borders; category III for tumor > 3 cm involving two levels (levels according to the ESGS proposal [8]); category IV for tumor > 3 cm involving more than 2 levels [9]. For instance, they reserve ECD only for pleomorphic adenomas of 3 cm or less that are mobile and close to the parotid borders (category I) and for category I and II of Warthin’s tumors.

We describe our outcomes in terms of facial palsy and local recurrences with ECD by utilizing the capsule margin as the surgical plane of dissection in the treatment of benign parotid tumors independently of their diameter.

## 2. Materials and Methods

This is a retrospective observational study evaluating all extracapsular dissection procedures for benign parotid gland tumors performed by two experienced (G.M.; D.T.) ear nose and throat (ENT) surgeons of Vanvitelli University between 2009 and 2017. 

The tumors were preoperatively assessed by ultrasound, computed tomography (CT), magnetic resonance imaging (MRI), or fine-needle aspiration (FNA). 

We included only benign tumors definitively confirmed by surgical histopathology in the study cohort. Patients presenting with recurrent lesions or a history of surgery on the affected parotid gland were excluded.

Imaging was reviewed to categorize the masses preoperatively, according to the classification scheme proposed by Quer [9]: category I, tumor ≤ 3 cm, superficial, mobile, and close to parotid borders; category II, ≤3 cm, deep, or far from borders; category III, >3 cm, involving two levels; category IV, >3 cm, involving >2 levels (levels according to the ESGS proposal [8]). Tumors involving levels III and IV have been classified as deep [8]. The size of the tumors was measured by ultrasound.

All included patients underwent ECD under facial nerve (FN) monitoring technology (NIM 3.0 Neuro, Medtronic). We used 4-channel configuration with the application of four subcutaneous electrodes at level of the muscles: frontal, orbicular of eye, orbicular of mouth, mental.

All procedures were conducted under general anesthesia.

Any documented sign of FN dysfunction, according to the House–Brackmann score [19], in the immediate postoperative period was considered FN weakness. The House–Brackmann classification score includes six grades: 

Grade I: Normal; 

Grade II: Slight facial weakness or other mild dysfunction, normal tone and symmetry at rest; complete closure of the eye without effort; slight asymmetry of the mouth when facial movements occur; 

Grade III: Assigned to patients dealing with moderate dysfunction. These patients generally do not display any noticeable facial weakness with synkinesis and can maintain complete eye closure and good forehead movement with effort. 

Grade IV: Assigned to patients dealing with severe dysfunction. Obvious facial weakness. Incomplete eye closure, no forehead movement, asymmetrical mouth movement, and synkinesis.

Grade V: Assigned to patients who have little to no ability to smile, frown, or make other facial expressions. The closure of the eye is incomplete, and there is no forehead movement.

Grade VI: No facial motion.

Facial nerve weakness was further characterized by using a threshold of more than 6 months of presence of the signs to distinguish persistent conditions from transient conditions. 

Follow up was at least 3 years (mean 49.29; SD 13.17). Each patient underwent echography annually to assess recurrences.

### 2.1. Surgical Technique

A preauricular incision follows the insertion of the earlobe and ascends towards the mastoid region. A second incision, perpendicular to the first, originates from the mastoid region, extending into the neck for a few centimeters along the anterior edge of the sternocleidomastoid muscle. Its course gives the incision a y-shape. 

After incising the superficial layers, including skin and subcutaneous tissue, anterior flaps were elevated. The posterior parotid border is isolated; the greater auricular nerve branches were exposed and usually preserved. 

We then incise the parotid along its posterior edge until we reach the tumor capsule. The isolation of the neoplasm can be carried out easily along its capsule, strictly respecting its integrity and without exposing the main trunk of the facial nerve. We used the capsule margin as the plane of dissection and identified only the facial nerve branches that were close to the tumor. Facial nerve monitoring was used in 100% of cases for identification of the facial nerve in the portion of the gland affected by the neoplasm. There was no case in which a reconstruction at the surgical site was performed. All surgical specimens were subjected to a histological examination of the tumor. Prior to sectioning, the entire core-resection was marked with ink for histological visualization of the excision margin.

### 2.2. Statistical Analysis

Descriptive analysis and a chi-squared test was performed with Microsoft Excel (Microsoft Office Professional Plus, 2016).

## 3. Results

A total of 88 extracapsular dissections that met inclusion criteria were performed for benign parotid tumors at the ENT Unit of Vanvitelli University between 2007 and 2017. We enrolled 57 (64.3%) females and 31 (35.7%) males; the mean age was 42.1 years (SD 15.9; range 21–65). 

The mean lesion size was 4.26 cm (SD 2.53; range 1.5–10). There were 68 tumors less than 3 cm in diameter and 20 that were larger, 64 tumors that were superficial, and 24 that were deep according to the ESGS classification [8]. 

According to the histological analysis, the most common lesion histotypes were pleomorphic adenoma (PA; 78 cases, 88.6%) and Warthin tumor (6 cases, 6.84%). Less common lesions included one case (1.14%) of myoepithelial tumor, one case (1.14%) of basal cell adenoma, and two cases (2.28%) of lipoma. 

Transient FN weakness (all resolved in less than 3 months) were found in four cases (4.54%) of PA, all with involvement of the marginal mandibular branch (House–Brackmann grade II) as a surgical complication.

Frey Syndrome and sialoceles were never observed. We did not observe any local recurrence at 3 years of follow up. 

We performed ECD in 24 cases with involvement of the deep lobe (20 cases of pleomorphic adenoma and 4 case of Warthin’s tumor) and in 20 cases of tumor size > 3 cm (Categories III and IV by Quer [9]), of which 18 were pleomorphic adenomas and 2 were Warthin’s tumors, without significant difference in complication rates between different sizes of tumor (≤3 versus >3 cm, *p* = 0.9) and locations (superficial versus deep, *p* = 0.91). The results are resumed in Table 1, Table 2 and Table 3 and Figure 1, Figure 2 and Figure 3.

## 4. Discussion

Although the extension of the resection in the treatment of benign parotid tumors ranged from limited resections to total parotidectomy with preservation of the facial nerve [9,20,21,22], there is still a tendency to use limited surgeries, especially in small benign tumors [7,8,9,11,12]. 

The surgery procedure aims to obtain the best possible results in terms of complete removal of the tumor, risk of recurrence, and postoperative outcome with preserved FN function. The rate of temporary postoperative paresis of the facial nerve is reported in the literature from 15% to 25% after superficial parotidectomy and 20% to 50% after total parotidectomy, whereas the rate of permanent facial nerve paresis is reported from 5% to 10% [22,23,24,25,26].

ECD has been offered as an alternative method to minimize morbidity following parotidectomy. According to the literature, ECD involves a careful dissection around the tumor capsule with a rim of 1–2 mm of normal glandular tissue, and without pre-identification of the FN [10]. Previous studies suggest that ECD has lower rates of complications without a higher recurrence rate in comparison to Superficial Parotidectomy (SP) [10,27,28]. 

Hancock [29,30] described local capsular dissection (LCD), which was intended as a very careful dissection along the capsule to ensure inclusion of possible lobular extensions. In 1993, Dallera [31] describes the results on 71 primary parotid pleomorphic adenomas undergoing LCD, which included a 5.6% recurrence rate.

A meta-analysis by Albergotti et al., as well as several other studies [10,27,28], report similar rates of recurrence in both ECD and SP, but with reduced rates of FN paresis and Frey Syndrome in favor of ECD. In addition, an updated meta-analysis by Xie et al. [12] indicated that ECD was a safer alternative surgical technique to treat selected small, superficial, and mobile benign lesions without adhering to FN.

Barzan and Pin [32] found that the incidence of permanent facial nerve damage was 1.3% for ECD and 4% for superficial parotidectomy. Dell’Aversana et al. demonstrated permanent facial nerve damage of 0% for ECD compared to 8.9% following superficial parotidectomy [28].

A recently published clinical trial has concluded that more functional surgery in benign tumors enables better cosmetic results, with better sensitivity, less morbidity, and equal tumor control [33].

The need to report our experience in this work must be seen in the light of Quer’s proposal (2017) to reserve the ECD, in cases of pleomorphic adenomas, only for tumors that are smaller than 3 cm and superficial. 

In 2017, Quer et al. [9] considered several parameters, such as size, site, and histopathology, to decide the extent of surgery in the treatment of benign parotid tumors and proposed establishing a four-category classification by choosing a cut-off of 3 cm in diameter as the cut-off. They reserve ECD only for pleomorphic adenoma that are 3 cm or less, mobile, and close to the parotid borders (category I) and for categories I and II of Warthin’s tumors. 

Auger’s study [20], in 2021, is the first to use the Quer classification to describe the functional outcomes of ECD by documenting its effectiveness even in cases of larger and deeper tumors.

We used ECD in our study population with the aim to reduce the incidence of tumor recurrence and complications; we performed ECD even in larger and deeper tumors. In our procedure, the capsule margin represents the plane of dissection, as evidenced by the histological examination (Figure 3). Facial nerve monitoring allowed us to identify the branches of the facial nerve at the site of the neoplasm, which minimized the risk of damage. In comparison to partial parotidectomy, which involves the removal of 1–2 cm of healthy parotid tissue, this procedure minimizes complications such as FN dysfunction and Frey syndrome. 

The use of the extracapsular margin as a dissection plane, regardless of the resection of 1–2 mm of macroscopically healthy tissue, allowed us to minimize the risk of damage of the FN due to the blunt dissection, even in cases of deep lobe involvement.

In this regard, Witt’s 2002 study reported that focal capsular exposure occurs in virtually all parotid surgery for PA independently of the surgical technique (Total Parotidectomy, Partial Superficial Parotidectomy, and Extracapsular Dissection) [22]. It is the capsular rupture that determines a significantly higher rate of recurrence, which did not vary among surgical approaches [22]. He reported that there were only 1/20 cases of capsular rupture in the ECD group without recurrences at 3 years of follow up [22]. In our series, capsular rupture occurred in one case, but no recurrence occurred at 3 years of follow up.

As reported in the literature, the time required for recurrent mixed tumors to clinically manifest varies from two to ten years after the first surgery [34,35,36]. Hancock, in 1999, did not report recurrences on a series of 28 patients who underwent ECD with an average follow-up of 10.3 years [34]. Martis, in 1983, reported 0 recurrences in 98 patients at two years of follow up [35]. Uyar, in 2011, did not report relapses with an average follow up of 194 months [36]. There were no recurrences during our study period (mean 49.29 months). Longer follow-up is certainly necessary for a definitive assessment of recurrence rate of this technique. No incidence of permanent FN weakness was observed in our cases, whereas the incidence of transient (lasting < 6 months) FN weaknesses were found only in four cases (4.54%) of PA with involvement of the marginal mandibular branch (House–Brackmann grade II) as a surgical complication. Facial nerve monitoring with electrical nerve stimulation and evaluation of evoked muscle potentials are of great use for mapping the surgical field when looking for the nerve–tumor dissection plan. Auger [20], in a similar case series, reported three cases of transient facial weakness in 20 patients undergoing ECD with partial facial dissection for tumors greater than 3 cm in diameter. 

We performed ECD in 24 cases with involvement of the deep lobe (20 cases of pleomorphic adenoma and 4 case of Warthin’s tumor) and in 20 cases of tumor size > 3 cm (Categories III and IV by Quer9), of which 18 were pleomorphic adenomas and 2 were Warthin’s tumors, without significant difference in complication rates between different sizes of tumor (≤3 versus >3 cm, *p =* 0.9) and locations (superficial versus deep, *p* = 0.91).

Witt [18], in 2016, reports with his technique of ECD with FN dissection led to 0% permanent facial nerve dysfunction (0 of 108 patients) and 4% minimal transient lower facial nerve dysfunction. The author compared these results with his previously published series of Partial superficial parotidectomy in which the rate of transient facial weakness was 17% [37]. McGurk [27] reported Frey’s syndrome and transient facial nerve palsy in the ECD of 5% and 10% of patients, respectively. Albergotti’s meta-analysis [10] of nine studies with 1,882 patients showed a rate of transient facial nerve paralysis of 8% and a symptomatic Frey’s syndrome in 4.5% of all cases for ECD. Dell’Aversana Orabona et al. [28], show transient facial weakness in 3.9% of cases with the ECD technique. In our experience, no reconstruction at the surgical site was performed [20,38].

## 5. Conclusions

Our results suggest that ECD with FN monitoring could be considered a suitable option for first-time diagnosed benign parotid tumors, even in cases of large dimension and deep lobe involvement, that could significantly reduce tumor recurrence and facial weakness. 

## Figures and Tables

**Figure 1 jpm-12-01641-f001:**
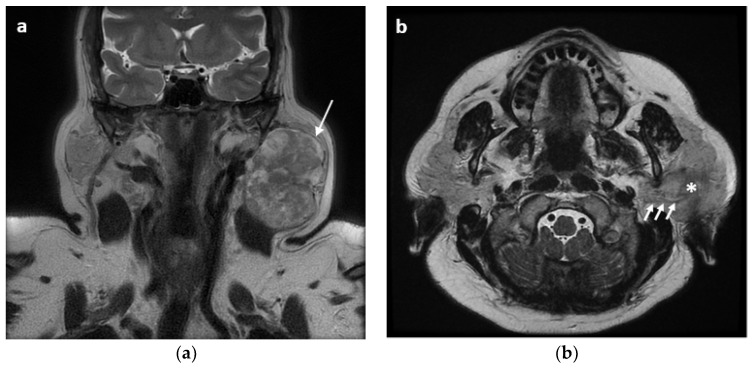
Pleomorphic adenoma of the left parotid gland in a 40-year-old woman. (**a**) Coronal T2-weighted MR image shows a heterogeneously hyperintense mass with the entire capsule (arrow) in the left parotid gland. (**b**) Axial T2-weighted MR image showing the neoplasm (asterisk) with adhesion to facial nerve (arrows).

**Figure 2 jpm-12-01641-f002:**
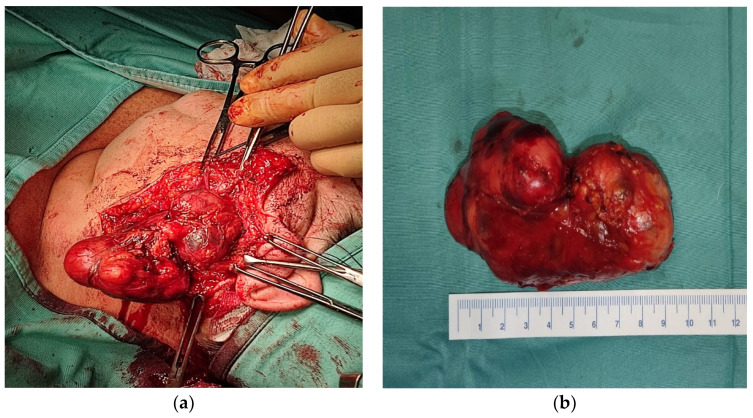
(**a**) Surgical field with the capsule of the tumor exposed and facial nerve preserved; (**b**) 10 cm parotid mass excised with intact capsule.

**Figure 3 jpm-12-01641-f003:**
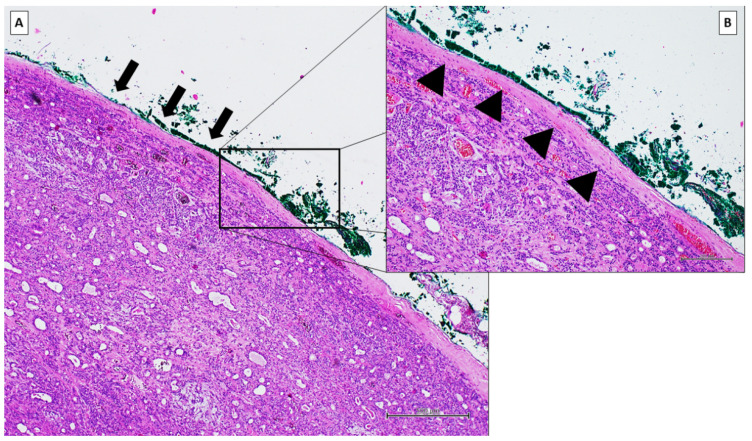
Representative histological examination of ECD of pleomorphic adenoma. (**A**) pleomorphic adenoma with the excision margin marked with ink (black arrow) (H&E stain, 10× magnification, scale-bar 500 µm). (**B**) detail of the ECD with the integrity of the capsule of the neoplasm histologically observed (black arrowhead) (H&E stain, 20× magnification, scale-bar 200 µm).

**Table 1 jpm-12-01641-t001:** Resumed descriptive results.

N. of patients		88
Age (mean)		42.1 y (SD 15.9; range 21–65)
sex	M	31
	F	57
Tumor size (mean)		4.26 cm (SD 2,53; range 1.5–10)
Depth	superficial	64 (54 category I + 10 category III)
	deep	24 (14 category II + 2 category III + 8 category IV)
Categories [9]	I	54
	II	14
	III	12
	IV	8
Istology	Pleomorphic Adenoma	68
	Warthin tumor	16
	Mioepitelioma	1
	Basal cells Adenoma	1
	Lipoma	2
complications	None	84
	Transient Facial nerve weakness (resolution within 3 months)	4 (4.54%)
	Permanent Facial nerve weakness	0
	Frey’s Sindrome	0
	sialocele	0

**Table 2 jpm-12-01641-t002:** Categories of patients according to Quer et al. [9].

Categories [9]	N. of Patients	%
I	54/88	61.4
II	14/88	15.9
III	12/88	13.6
IV	8/88	9.1

**Table 3 jpm-12-01641-t003:** Complications according to size and categories (Categories according to Quer et al. [9]).

Tumor Size	≤3 cm	>3 cm
Total of patients n.	68	20
Facial weakness transient	3/68 (4,41%)	1/20 (5%)
Facial weakness transient (according to categories [9])		
I	2/54	
II	1/14	
III		1/12
IV		0/8

## Data Availability

Data are available upon reasonable request.

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
