# Peer review of "Minimal Margin Surgery and Intraoperative Neuromonitoring in Benign Parotid Gland Tumors: Retrospective Clinical Study"

_jpm, 2022, doi:10.3390/jpm12101641_

Round 1

Reviewer 1 Report

There is an important and interesting publication. We are constantly looking for solutions that are effective, but not very invasive. The article certainly meets the expectations of the ENT doctors, however in the conclusions, that sound good I would have to add that the observation is only at the level of 3 - 4 years, not longer.

Author Response

  • Given the low long-term recurrence rate for pleomorphic adenoma, there is currently a trend towards minimally invasive surgery. As reported in the Literature, the time required for recurrent mixed tumors to be clinically manifest varies from two to ten years after the first surgery. Hancock, in 1999, on a series of 28 patients who underwent ECD, did not report recurrences, with an average follow-up of 10.3 years. Martis, in 1983, reported 0 recurrences in 98 patients at two years of follow up. Uyar, in 2011, did not report recurrences with an average follow up of 194 months. There were no recurrences during our study period (mean 49.29 months). Longer follow-up is certainly necessary for a definitive assessment of recurrence rate of this technique. [Hancock BD. Clinically benign parotid tumours: local dissection as an alternative to superficial parotidectomy in selected cases. Ann R Coll Surg Engl 1999;81:299–301; Martis C. Parotid benign tumors: comments on surgical treatment of 263 cases. Int J Oral Surg 1983;12:211–220;Uyar Y, Caglak F, Keles B, Yildirim G, Salturk Z. Extracapsular dissection versus superficial parotidectomy in pleomorphic adenomas of the parotid gland. Kulak Burun Bogaz Ihtis Derg 2011;21:76–79.]

Reviewer 2 Report

The topic of this article is one o interest for the reader. The hope of results with minimal functional and aesthetic disturbance in the same time with no recurrence is a continuous concern for the head and neck surgeon. 

Some questions need an answer.

In the studied series, more of the patients have more than 3 years since was operated. In my opinion, it is very to write what was happened with this patients in term of recurrence , knowing that as the time pass, as the chance for the recurrence increase.

It is hard to believe that, especially for pleomorfic adenoma, ECD a close dissection to the capsule the free of tumour limits is always  obtained. From this point of view, is very important in how many situations, the pathologist find situation where  the limits of the dissection is involved .

Knowing that elongation of the facial nerve itself produce functional embarrassment , how can the authors explain the exceptional results also for the bigger than 3 cm deep  tumour.

The FNA has not 100% certitude . In such a great series of patients, the authors has no switch of the diagnostic between FNA and final surgical sample examination, especially toward malignancy. If it is the case, what was th attitude?

Author Response

  • Given the low long-term recurrence rate for pleomorphic adenoma, there is currently a trend towards minimally invasive surgery. As reported in the Literature, the time required for recurrent mixed tumors to be clinically manifest varies from two to ten years after the first surgery. Hancock, in 1999, on a series of 28 patients who underwent ECD, did not report recurrences, with an average follow-up of 10.3 years. Martis, in 1983, reported 0 recurrences in 98 patients at two years of follow up. Uyar, in 2011, did not report recurrences with an average follow up of 194 months. There were no recurrences during our study period (mean 49.29 months). Longer follow-up is certainly necessary for a definitive assessment of recurrence rate of this technique. [Hancock BD. Clinically benign parotid tumours: local dissection as an alternative to superficial parotidectomy in selected cases. Ann R Coll Surg Engl 1999;81:299–301; Martis C. Parotid benign tumors: comments on surgical treatment of 263 cases. Int J Oral Surg 1983;12:211–220;Uyar Y, Caglak F, Keles B, Yildirim G, Salturk Z. Extracapsular dissection versus superficial parotidectomy in pleomorphic adenomas of the parotid gland. Kulak Burun Bogaz Ihtis Derg 2011;21:76–79.]
  • In our cases, experienced surgeons use the extracapsular margin as a dissection plane, regardless of the resection of 1-2 mm of macroscopically healthy tissue. The tumor capsule is then exposed in all cases. Capsular rupture occurred in one case but no recurrence occurred at 3 years of follow up. In this regard, Witt reports, in his 2002 study, that focal capsular exposure occurs in virtually all parotid surgery for PA, independently of the surgical technique (Total Parotidectomy, Partial Superficial Parotidectomy, and Extracapsular Dissection). It is the capsular rupture that determines a significantly higher rate of recurrence and did not vary among surgical approaches. He reports 1/20 cases of capsular rupture in ECD group without recurrences at 3 years of follow up. [Witt RL The significance of the margin in parotid surgery for pleomorphic adenoma. Laryngoscope, 2002; 112:2141–2154]
  • The use of the extracapsular margin as a dissection plane, regardless of the resection of 1-2 mm of macroscopically healthy tissue, allows to minimize the risk of damage of the FN, due to the blunt dissection. Facial nerve monitoring with electrical nerve stimulation and evaluation of evoked muscle potentials are of great use for mapping the surgical field looking for the nerve-tumor dissection plan. Auger, in a similar case series, reports three cases of transient facial weakness on 20 patients undergoing ECD with partial facial dissection for tumors greater than 3 cm in diameter. [Auger SR, Kramer DE, Hardy B, Jandali D, Stenson K, Kocak M, Al-Khudari S. Functional outcomes after extracapsular dissection with partial facial nerve dissection for small and large parotid neoplasms. Am J Otolaryngol. 2021;42(1):102770.]
  • In our series, we retrospectively selected only patients with final histological report of benignity. In these cases, all FNAC performed indicated absence of malignant cells.